# Baculovirus-Mediated Gene Therapy: Targeting BIRC6 for Lung and Breast Cancer

**DOI:** 10.3390/v17111458

**Published:** 2025-10-31

**Authors:** Abril Marchesini, Santiago M. Gómez Bergna, Leslie C. Amorós Morales, María Florencia López, Larisa Vásquez, Silvana E. Tongiani, Florencia González Morán, Víctor Romanowski, María Florencia Gottardo, Matias L. Pidre

**Affiliations:** 1Instituto de Biotecnología y Biología Molecular (IBBM, UNLP-CONICET), Facultad de Ciencias Exactas, Universidad Nacional de La Plata, Consejo Nacional de Investigaciones Científicas y Técnicas, La Plata B1900, Argentina; abrilmarchesini@biol.unlp.edu.ar (A.M.); sgomezbergna@biol.unlp.edu.ar (S.M.G.B.); amorosleslie@biol.unlp.edu.ar (L.C.A.M.); mfl@biol.unlp.edu.ar (M.F.L.); larisavasquez@biol.unlp.edu.ar (L.V.); stongiani@biol.unlp.edu.ar (S.E.T.); victor@biol.unlp.edu.ar (V.R.); 2Centro de Oncología Molecular y Traslacional, Departamento de Ciencia y Tecnología, Universidad Nacional de Quilmes, Bernal B1876BXD, Buenos Aires, Argentina; gonzalezmoranflorencia@gmail.com (F.G.M.); florenciagottardo@gmail.com (M.F.G.)

**Keywords:** BIRC6, baculovirus, shRNA, lung cancer, breast cancer, apoptosis

## Abstract

BIRC6, a member of the inhibitor of apoptosis protein family (IAP), regulates apoptosis, autophagy and cytokinesis. IAPs are often overexpressed in tumors, contributing to oncogenesis, therapy resistance and worse prognosis. In particular, BIRC6 overexpression has been found in several tumor tissues. The aim of this study was to evaluate the effect of BIRC6 silencing on the apoptotic response of breast and lung tumor cells. We used RNA interference based on short hairpin RNA (shRNA) to knock down gene expression encoded by a recombinant baculovirus (BV), an insect-specific virus unable to replicate in mammalian hosts, to carry out preclinical validation tests in experimental models both in vitro and in vivo. Our results indicate that BIRC6 plays an antiapoptotic role in both breast and lung tumor cells. In vivo, treatment with BV-shBRIC6 reduced breast and lung tumor progression and increased overall survival. After histological analysis, BV-shBRIC6 was able to increase tumor necrosis. In addition, we demonstrated that BIRC6 expression correlates with antiapoptotic and tumor progression-relevant markers in lung and breast cancer patients. BV-based silencing of BIRC6 may have therapeutic value for the treatment of lung and breast tumors. Further translational studies of BV-shBIRC6 in lung and breast cancer are warranted.

## 1. Introduction

Baculoviruses are insect-specific DNA viruses that infect arthropods, mainly within the order Lepidoptera [1]. Within the Baculoviridae family, Autographa californica multiple nucleopolyhedrovirus (AcMNPV) is the most studied. Although baculoviruses are insect-specific pathogens; they are able to transduce mammalian cells, allowing transgene delivery and expression without viral replication nor genome integration. These features have made baculoviruses a promising vector for gene therapy application [2,3,4,5]. In comparison with other viral vectors used for gene therapy, baculoviruses have a large transgene capacity, allowing more than 38 kbp of foreign material, and are easily propagated by infecting an insect cell monolayer and harvesting the supernatant of infection two or three days later. Unless it is required by the type of transgen carried on the genome, baculoviruses can be manipulated in a biosafety level 1 lab (BSL-1), with no need for special equipment [2]. In recent years, baculoviruses have been widely used for foreign gene delivery into mammalian cells and in particular for the treatment of different tumors such as pituitary tumors [6], ovarian tumors [7] and CNS tumors [8,9]. Altogether, these approaches have shown that baculoviruses are an important tool for gene therapy.

Apoptosis is a type of programmed cell death (PCD) highly related to the control of cell population and its alteration, and the consequent loss of balance between cell division and cell death can lead to pathological conditions. More precisely, in the context of a neoplastic disease, some cancer cells can avoid apoptosis to contribute to tumor progression. Because of this, recent anticancer strategies are based on the targeting of IAPs (inhibitors of apoptosis proteins) to reactivate cell death. IAPs contain highly conserved interaction motifs called baculoviral IAP repeats (BIRs) and mostly a ubiquitin ligase domain and are involved in several biological activities that promote cancer cell survival and proliferation [10].

Breast cancer (BC) is one of the most common cancers among women worldwide. It can be classified by molecular and histological characteristics into three groups: BC expressing hormone receptor (estrogen receptor (ER+) or progesterone receptor (PR+), BC expressing human epidermal receptor 2 (HER2+) and triple-negative breast cancer (TNBC) (ER−, PR−, HER2−) [11], with TNBC being the most invasive [12]. The mechanism of occurrence of breast cancer is still controversial; however, it is well known that apoptosis plays a vital role in BC appearance and progression [13].

On the other hand, lung cancer is the leading cause of cancer-related death and the second most frequently diagnosed cancer worldwide (Globocan). Lung cancer comprises two main morphological groups: small cell lung cancer (SCLC, 15% of cases) and non-small cell lung cancer (NSCLC, 85% of cases). Within NSCLC classifications, with adenocarcinoma (LUAD) being the most common subtype, accounting for more than 40% of all cases, followed by squamous cell carcinomas (20%). Despite advances in its diagnosis and treatment, the survival rate of patients remains low (15% within a maximum period of 5 years) [14,15].

BIRC6 (baculoviral IAP repeat containing 6, also known as Apollon or BRUCE in mice) is a giant member of the IAP family with approximately 530 kDa [16,17].

This protein acts as a ubiquitin ligase and facilitates proteasomal degradation of SMAC and caspase-9, among others. On the other hand, BIRC6 plays a role as an antiapoptotic IAP by inhibiting the proapoptotic protein SMAC, preventing the cleavage of procaspase-9 or by binding to active caspases, including caspases 3, 6, 7 and 9, through its BIR domain [17,18,19,20,21].

BIRC6 has been related to the development and progression of different types of cancer in humans. Its antiapoptotic activity favors the evasion of cell death, while its involvement in the DNA damage response contributes to resistance to conventional therapies [22,23,24]. In addition, its dual role in autophagy facilitates metabolic adaptation under stress conditions [25,26,27]. Consistently, BIRC6 overexpression has been reported in several cancer types, including triple-negative breast cancer, prostate, renal, colorectal, childhood acute myeloid leukemia, ovarian, hepatocellular, esophageal squamous cell carcinoma, and non-small cell lung cancer (NSCLC) [28,29,30,31,32,33,34,35]. In most of these cases, BIRC6 expression has been shown to correlate with carcinogenesis, tumor progression and poor patient prognosis and has been shown to enhance chemoresistance [29,30].

In NSCLC, high BIRC6 expression has been proposed as a marker to identify stage I patients at risk of relapse after surgery, who could benefit from adjuvant therapies. Moreover, BIRC6 contributes to malignant progression by conferring resistance to chemotherapy, positioning it as a promising therapeutic target to sensitize tumor cells and improve treatment options [29,30].

In breast cancer, BIRC6 has been suggested to play a key role in tumor homeostasis and in different stages of the metastatic cascade, including angiogenesis, migration and adhesion [36]. In particular, in TNBC, BIRC6 expression was found to be elevated and associated with poorer survival, in addition to positively correlating with the expression of epidermal growth factor receptor (EGFR) in cells and tissues [28].

Therefore, in this work, we present a new baculovirus-based gene therapy vector designed to silence the expression of the inhibitor of apoptosis BIRC6 and its preclinical validation on lung and breast cancer experimental models.

## 2. Materials and Methods

Reagents were purchased from Sigma-Aldrich (Merck KGaA, Darmstadt, Alemania) except for Eagle’s medium (MEM; GIBCO, Thermo, Waltham, MA, USA) and supplements such as fetal bovine serum (Internegocios, Buenos Aires, Argentina). TUNEL reagents were obtained from Promega (Madison, WI, USA), antibodies from Vector Laboratories Inc. (Burlingame, CA, USA), plasmids from GenScript Inc. (Piscataway, NJ, USA) and primers from Macrogen (Seoul, South Korea).

### 2.1. Plasmids Encoding birc6-Specific Short Hairpin RNA

The 21-nucleotide-long shRNA coding dsDNA against the BIRC6 mRNA sequence was designed following the guidelines published by Bofill-De Ros [37]. Then, three shRNAs were synthesized by GenScript and cloned under the control of the U6 promoter into the pBacPak9 transfer vector. In addition, the coding sequence for dTomato was also included under the control of the cytomegalovirus (CMV) major immediate-early (IE) promoter in order to identify transduced cells (Figure 1C).

### 2.2. Generation of Recombinant Baculoviruses

For in vitro and in vivo experiments, recombinant AcMNPV baculoviruses were produced containing the complete cassette as previously described. AcMNPV sequences present in the pBacPak9 backbone allow homologous recombination in insect cells, between the transfer vector and the viral DNA. Then, recombinant pBacPAK9 was co-transfected in the insect cell line High Five^TM^ (Thermo, Waltham, MA, USA) with the occlusion positive (occ+) bacmid DNA (bApGOZA) [38,39,40]. High Five^TM^ cells were cultured using Grace’s medium (Thermo, Waltham, MA, USA), supplemented with 10% of fetal bovine serum in T-flasks at 27 °C until signs of infection (polyhedron) became apparent, producing BV-shBIRC6. A recombinant baculovirus expressing only dTomato was generated as a control (BV-Control) using the PluriBAC system as we reported previously [41]. Recombinant dTomato expression was detected by microscopy (Nikon Eclipse Ti-S, Nikon Instruments Inc. Melville, NY, USA). BVs were titrated on High Five^TM^ cell monolayers as plaque forming units (PFUs).

### 2.3. Tumor Cell Cultures

Human lung adenocarcinoma cell line A549 (ATCC, CCL-185™), mice lung cancer cell line 3LL (RRID:CVCL_5653), human triple-negative breast cancer cell line MDA-MB-231 (ATCC, HTB-26™) and mice breast cancer cell line F3II [42,43] were used in this study. Cell monolayers were grown with MEM medium supplemented with 10% fetal bovine serum. Then, they were harvested using 0.025% trypsin-EDTA in PBS. Cell viability was assessed by trypan blue exclusion. In vitro experiments: for the TUNEL assay, cells were seeded on cover slips placed in 24-well tissue culture plates (1 × 10^5^ cells·mL^−1^·well^−1^); for the immunocytochemical assay, cells were seeded in 24-well tissue culture plates (2 × 10^5^ cells·ml^−1^·well^−1^); finally, for the RNA extraction, cells were seeded in 12-well tissue culture plates (5 × 10^5^ cells·ml^−1^·well^−1^).

### 2.4. BV-Based Gene Transduction

Tumor cells were incubated with recombinant baculoviruses for 2 h with 2 × 10^9^ PFUs of the recombinant baculoviruses in PBS. After the addition of supplemented MEM medium, they were incubated for 48 or 72 h and fixed using 4% paraformaldehyde (PFA) for BIRC6 detection by immunofluorescence and for the assessment of apoptosis by TUNEL. Another set of transduced cells were processed for RNA extraction.

### 2.5. Expression of BIRC6 by Immunofluorescence

The presence of BIRC6 in tumor cells was evaluated by indirect immunofluorescent staining. Cells were fixed with PFA 4% for 10 min. Permeabilization was performed using BD Fix and Perm Buffer (BD Biosciences, San Jose, CA, USA) for 5–10 min. Cells were incubated with a blocking solution containing PBS, 0.05% Tween-20, and 5% FBS for 40 min at room temperature. For BIRC6 detection, cells were incubated for 1 h in a wet chamber with anti-BIRC6 antibody (Sigma Aldrich Merck KGaA, Darmstadt, Alemania, 1:100 PBS-BSA 2.5%), washed and incubated for 30 min with anti-rabbit IgG-Alexa488 (Vector Laboratories Inc., Newark, CA, USA, 1:100) at room temperature and in the dark. The cell’s nuclei were stained with DAPI 1:10,000, and finally, preparations were mounted using Mowiol 4-88 (Merck, Darmstadt, Germany). Control slides were incubated with the corresponding secondary antibody. Cells were visualized using a fluorescence light microscope (Nikon Eclipse Ti-S, Nikon Instruments Inc. Melville, NY, USA).

### 2.6. Expression of BIRC6 by Flow Cytometry

The expression of BIRC6 in tumor cells was evaluated by flow cytometry. Cells were fixed with PFA 4% for 10 min. Permeabilization was performed using BD Fix and Perm Buffer (BD Biosciences, San Jose, CA, USA) for 5–10 min. Cells were incubated with a blocking solution containing PBS, 0.05% Tween-20, and 5% FBS for 40 min at room temperature. For BIRC6 detection, cells were incubated for 1 h in a wet chamber with anti-BIRC6 antibody (Sigma Aldrich, Merck KGaA, Darmstadt, Alemania, 1:100 PBS-BSA 2.5%), washed and incubated for 30 min with anti-rabbit IgG-Alexa488 (Vector Laboratories Inc., Burlingame, CA, USA; 1:100 PBS-BSA 2.5%) at room temperature and in the dark. Cells after staining were analyzed by flow cytometry (FACS) in a FACScalibur device (Franklin Lakes, NJ, USA). To determine the cut-off point for Alexa488 fluorescence, unstained cells were used. Data obtained by flow cytometry was analyzed with FlowJo software v10 program.

### 2.7. Microscopic Detection of DNA Fragmentation by TUNEL

At 48 or 72 h post-transduction, the culture medium was removed and the wells were washed with 1× phosphate-buffered saline (PBS). Coverslips were fixed with 4% paraformaldehyde (PFA) for 25 min at 4 °C, followed by two 5 min washes with 1× PBS. For cell permeabilization, samples were incubated with 0.2% Triton X-100 for 5 min and then washed twice more with 1× PBS for 5 min each. TUNEL staining was performed following the manufacturer’s instructions of the DeadEndTM Fluorometric TUNEL System (Promega, Madison, WI, USA). For this, 100 µL of equilibration buffer was added to each well and incubated at room temperature for 5–10 min. Then, 50 µL of the reaction mix containing the TdT enzyme was added, and the samples were incubated for 60 min at 37 °C in a humidified chamber, protected from light from this step onward. The reaction was stopped by incubating the samples in 2× SSC for 15 min. Coverslips were then washed three times with 1× PBS for 5 min, and the cell’s nuclei were stained with DAPI 1:10,000; finally, preparations were mounted using Mowiol 4-88 (Merck, Darmstadt, Germany). Cells were visualized in a fluorescence light microscope (Nikon Eclipse Ti-S, Nikon Instruments Inc., Melville, NY, USA). Data of apoptotic cells were expressed as the percentage of cultured apoptotic cells ([(TUNEL+)/total cells] × 100).

### 2.8. RNA Isolation and RT-qPCR

Tumor cells were transduced with BV-shBIRC6 or BV-Control and incubated for 72 h. Transduced cells were collected as described above and RNA extraction was performed using Trizol reagent (Thermo, Waltham, MA, USA). One µg of total RNA was treated with DNAse (Promega) following the manufacturer’s instructions. After 1 h at 37 °C, STOP solution was added and reaction was incubated for 10 min at 65 °C. One µg of digested RNA was reverse transcribed using M-MLV reverse transcriptase according to the manufacturer’s protocol (Promega). RNA samples were incubated with 1 µL primers (10 uM) at 70 °C for 5 min, then transferred to ice immediately. A mix of 5 µL of M-MLV 5× reaction buffer, 5 µL of dNTP’s (10 mM), recombinant RNasin^®^ ribonuclease inhibitor (25 units), M-MLV RT (200 units) and nuclease-free water to a final volume of 25 µL was added to each sample, followed by 60 min at 42 °C.

For real-time RT-qPCR, two pairs of primers mapping to the BIRC6 and YWHAZ genes, encoding the 14-3-3 protein zeta/delta, respectively, were used (Table 1). All primers were obtained from Macrogen (Seoul, South Korea). Real-time PCR was performed using a HOT FIREPol^®^ EvaGreen^®^ qPCR Mix Plus (Solis BioDyne, Tartu, Estonia). A total of 10 µL of solution containing 1 µL of cDNA, 1 µL of forward and reverse primers (10 mM) and 2 µL of HOT FIREPol^®^ EvaGreen^®^ qPCR Mix Plus was used. All reactions were performed in triplicate. Negative controls included water as a template. Amplification cycle was initiated by a 12 min preincubation at 95 °C, followed by 40 cycles at 95 °C for 15 s, 58 °C for 20 s and 72 °C for 30 s, ending at 72 °C for 2 min. Gene expression was normalized using the reference gene YWHAZ by the ΔΔCt method and expressed as fold changes relative to the control group [44].

### 2.9. Animals

Female BALB/c or C57BL/6 mice were kept in accordance with the National Institutes of Health (NIH) Guide for the Care and Use of Laboratory Animals. Protocols were previously approved by the University of Quilmes Ethics Committee (Res. Nº 011/2015; 5 November 2015). Controlled conditions of light (12:12 h light–dark cycles) and temperature (20–25 °C) were employed. Mice were fed with standard laboratory chow and water ad libitum.

### 2.10. Breast Cancer Experimental Model

Eight-week-old female BALB/c mice were injected subcutaneously (s.c.) into the right flank with 2 × 10^5^ cells F3II cells as we reported previously [45]. When the tumor volume reached approximately 200 mm^3^, animals were randomized in two cages and individualized. Then, animals were injected intratumorally with 10^8^ PFUs (40 μL/mouse) of BV-Control (*n* = 5) or BV-shBIRC6 (*n* = 5).

Tumor size was measured every 48 h with a caliper, and tumor volume was estimated as [width^2^ × length]/2 (mm^3^). Tumor growth was determined during the entire protocol, and survival was evaluated until the tumor reached 2000 mm^3^.

Periodically, mice weight control was carried out, as well as toxicity, fur, behavior, food and drink, and when signs of distress appeared, they were euthanized by cervical dislocation.

Tumors were fixed in 4% PFA and embedded in paraffin. We deparaffinized 4 µm width sections in xylene, rehydrated them in ethanol and stained for determination of necrosis.

### 2.11. Lung Cancer Experimental Model

Eight-week-old female C57BL/6 mice were injected subcutaneously (s.c.) into the right flank with 3 × 10^6^ 3LL cells, as was reported previously by Segatori et al. [46]. When the tumor volume reached approximately 200 mm^3^, animals were randomized in two cages and individualized. Then, animals were injected intratumorally with 10^8^ PFUs (40 μL/mouse) of BV-Control (*n* = 6) or BV-shBIRC6 (*n* = 7).

Tumor size was measured every 48 h with a caliper, and tumor volume was estimated as [width^2^ × length]/2 (mm^3^). Tumor growth was determined during the entire protocol, and survival was evaluated until the tumor reached 2000 mm^3^.

Periodically, mice weight control was carried out, as well as toxicity, fur, behavior, food and drink, and when signs of distress appeared, they were euthanized by cervical dislocation.

Tumors were fixed in 4% PFA and embedded in paraffin. We deparaffinized 4 µm width sections in xylene, rehydrated them in ethanol and stained for determination of necrosis.

### 2.12. Tumor Necrosis Assessment

For tumor necrosis assessment, the workflow previously reported by Solernó et. al. [47] was applied. Brightfield images of whole H&E-stained tumor sections were captured at 2.5× magnification with a Cytation Gen5 Reader (BioTek, Winooski, VT, USA). Quantification of necrotic regions within the tumor tissue was carried out using ImageJ software version 1.5j8 (NIH, Bethesda, MD, USA, imagej.nih.gov).

Necrosis was defined as areas displaying intense eosinophilia, and both necrotic (NA) and viable (VA) areas were quantified employing the “Color Threshold’’ tool. Tumor necrotic rate (TNR) was then calculated as: “TNR  =  (NA × 100)/(VA  +  NA)” in 4 sections per experimental group. Adjusted tumor necrotic rate (ATNR) was calculated as: “ATNR  =  100 − (100 − TNR) × RTGR”, where RTGR stands for group-specific relative tumor growth rates.

### 2.13. Statistical Analysis

All the experiments were performed at least twice. Data are expressed as the mean ± SD and analyzed using GraphPad Prism (GraphPad Software, version 8.00) and R studio (2025.05.0+496). RT-qPCR data analysis was performed by Student’s *t*-test. TUNEL data from three different experiments are expressed as the percentage of TUNEL-positive cells with ±95% confidence limits (CLs) and analyzed by the χ2 test. Tumor growth was analyzed by multiple regression analysis. A log-rank test was used in order to evaluate Kaplan–Meyer survival curves. Differences between groups were considered significant when *p* < 0.05.

## 3. Results

### 3.1. Baculoviruses Efficiently Transduced Human and Murine Lung and Breast Cancer Cells

Firstly, we evaluated the capacity of BV-Control (Figure 1A), encoding the sequence of the fluorescent protein dTomato, to transduce either lung cancer cells, A549 and 3LL, or breast cancer cells lines, F3II and MDA-MB-231. For this, all the cell lines were incubated with 2 × 10^11^ PFUs of BV-Control for 72 h. dTomato-positive cells were detected by an epifluorescent microscope, and the cell nucleus was marked with DAPI (Figure 1B). As a result, BV-Control was able to transduce all the tumoral cell lines evaluated, with an efficiency of around 40% for 3LL, A549 and MDA-MB-231 cells, but less than 10% for F3II cells (Figure 1B).

### 3.2. Generation of BIRC6-Silencing Baculoviral Vector

As we mentioned in the Materials and Methods section, we designed and validated three different shRNAs against BIRC6 mRNA. Here, we present the results obtained with the very one that showed efficient silencing ability.

In order to evaluate the effects of BIRC6 silencing in lung and breast cancer cells, we designed one shRNA targeting BIRC6 mRNA (Figure 1C). We then generated BV-shBIRC6 recombinant baculoviruses via double homologous recombination in insect cells. To confirm the generation of the recombinant baculoviruses, cells were observed until signs of infection appeared, and dTomato expression was detected by an epifluorescent microscope (Figure 1D).

### 3.3. Effect of BIRC6 Silencing In Vitro

Once recombinant baculovirus was obtained, we evaluated if the designed shRNA carried by the BVs was able to efficiently silence birc6 expression in lung and breast cancer cells. With this aim, we evaluated silencing capacity using A549 cells as a model. A549 cells were treated with 2 × 10^11^ PFUs of BV-shBIRC6(1), (2) and (3), as well as with BV-Control. After 72 h of treatment, we observed that cells treated with BV-shBIRC6(3) show a decrease in BIRC6 detection (black arrow) compared with those treated with BV-Control by flow cytometry (Figure 2A).

Moreover, we confirmed BV-shBIRC(3) silencing capacity with a qRT-PCR assay. A549 cells were treated either with BV-shBIRC6 or BV-Control, and changes in mRNA levels were assessed at 72 h post-transduction. A549 cells transduced with BV-shBIRC6 showed a significant decrease in BIRC6 mRNA compared with the control group (Figure 2B). BIRC6 silencing was also detected by immunofluorescence in A549 and F3II cell lines (Figure 2C).

### 3.4. BIRC6 Silencing Induced Apoptosis In Vitro

Since BIRC6 is a member of the IAP family, we evaluated the effects in the apoptotic response in A549 cells treated with BV-shBIRC6(3) (Figure 3A). To accomplish this, A549 cells were incubated with BV-shBIRC6(3) and BV-Control for 72 h, and apoptosis induction was evaluated with a TUNEL assay. The BV-shBIRC6(3) treatment resulted in a significant increase in TUNEL-positive cells, around 35%, compared with BV-Control-treated cells after 72 h. These results suggest that the BIRC6 silencing mediated by our BV was effective enough to induce the apoptosis response in A549 lung cancer cells.

Additionally, we studied the effect of BIRC6 silencing over the apoptotic response in F3II breast cancer cells. For this, apoptosis induction was determined by the TUNEL assay (Figure 3B). Cells treated with BV-shBIRC6(3) presented a higher number of TUNEL-positive cells, approximately 5%, in comparison with BV-Control-treated cells. Suggesting that BIRC6 silencing mediated by BV-shBIRC6(3) is also efficient enough to induce an apoptotic response in the F3II breast cancer cell line.

### 3.5. Baculovirus-Mediated BIRC6 Silencing Reduced Tumor Growth and Increased Survival in Mice Experimental Models of Lung Cancer

To evaluate the effect of BIRC6 silencing on tumor growth, a murine experimental model with established tumors was generated and challenged with baculoviral transduction. To achieve this goal, C57BL/6 mice were injected with 3LL lung cancer cells. Once tumors grew up to a volume between 150 and 200 mm^3^, mice received an intratumoral injection of BV-shBIRC6(3) or BV-Control. Tumor size was measured from the day of inoculation, and survival was monitored until the tumor reached 2000 mm^3^ or showed necrosis symptoms (Figure 4A).

In mice treated with BV-shBIRC6(3), tumor growth was significantly delayed compared with animals treated with BV-Control (Figure 4B). In the same way, the tumor volume at the end of the experiment was significantly smaller in the group injected with BV-shBIRC6 in comparison with those treated with BV-Control (Figure 4C). Also, tumor growth rate was significantly higher in BV-Control compared with BV-shBIRC6(3)-treated mice (Figure 4D).

In addition to these results, the survival rate of mice treated with BV-shBIRC6(3) is also significantly higher compared with the control group (Figure 4E). Assessment of tumor necrosis in lung cancer has become a relevant tool for evaluating the response to different types of therapies [48,49,50], and adjusted tumor necrotic rates (ATNRs) were calculated (Figure 4F). BV-shBIRC6(3) was capable of significantly enhancing ATNRs in comparison with BV-Control-treated animals from 78.11  ±  4.88 to 58.36  ±  2.34%, respectively. Viable tumor tissue areas were found as large and scattered basophilic regions, interrupted by vast areas of necrotic tissue (Figure 4G).

### 3.6. BIRC6 Silencing Reduced Tumor Growth in Mice Experimental Models of Breast Cancer

As we did in the in vitro analysis, we also explored the effects of BIRC6 silencing in in vivo assays of breast cancer. BALB/c female mice were injected into the breast with F3II breast cancer cells. Once again, when tumors grew up to a volume between 150 and 200 mm^3^, mice received an intratumoral injection of BV-shBIRC6(3) or BV-Control. Tumoral size was measured from day 22 post-inoculation, and animals were sacrificed after 45 days (Figure 5A). In this case, not only tumoral growth was significantly delayed (Figure 5B), but also the tumor volume was significantly smaller in mice treated with BV-shBIRC6(3) compared with those injected with BV-Control (Figure 5C).

In addition, as we observed in our lung cancer in vivo experiment, breast tumor growth rate was significantly higher in BV-Control compared with BV-shBIRC6(3)-treated mice (Figure 5D). Once again, ATNRs were calculated for both experimental groups (Figure 5E). BV-shBIRC6(3) was able to significantly enhance ATNR in comparison with BV-Control-treated animals from 80.10 ± 4.77 to 40.86 ± 6.45%, respectively (Figure 5F).

## 4. Discussion

Baculoviruses have been utilized as expression vectors (BEVS) since the early 1980s. Initially, these viruses attracted interest due to their potential application as biological control agents, representing an environmentally friendly alternative to chemical insecticides in the agricultural sector. However, the successful expression of recombinant human interferon-β (IFN-β) [51] using a genetically engineered Autographa californica multiple nucleopolyhedrovirus (AcMNPV) in lepidopteran host cells marked a significant advancement. This milestone positioned BEVS as a robust and versatile platform for heterologous protein production. Since then, the system has facilitated the expression of thousands of recombinant proteins, several of which have progressed to commercial availability.

BEVS consists of a technological platform significantly safer than other viral vectors and usually requires the generation of at least one recombinant baculovirus. Most of the available baculovirus production systems are based on homologous recombination (HR). In particular, recombinant baculovirus generation systems have recently been developed that allow for the efficient construction of transfer vectors by modular Golden Gate assembly [41].

Owing to all these advantages, baculoviruses have been increasingly used as gene therapy vectors in several applications, including vaccine generation, cancer treatment and regenerative medicine [6,7,8,9,52,53,54,55]. Because of their versatility, baculoviruses could allow us to use either CRISPR/Cas9 technology to generate gene knockouts [56,57] or RNA interference approaches for post-transcriptional gene silencing [6,7,9]. To achieve our goal, we preferred RNA interference to avoid the generation of additional DNA damage in cell lines and models (tumoral) in which DNA repair pathways and DNA integrity checkpoints are often mutated.

Here, we have evaluated baculovirus-mediated transduction in multiple human and mice experimental models for breast cancer and lung cancer. We observed in vitro transduction, although with varying efficiencies between the different models. Baculovirus entry into mammalian cells has been shown to involve interactions with cell surface heparan sulfate proteoglycans, particularly syndecans, which facilitate viral binding and subsequent transduction [58,59]. It was proposed that endocytosis mediated by clathrin [60], in a low pH-dependent manner, is the primary mechanism of entry in mammalian cells. Moreover, it was observed that virus fusion in early endosomes seems to be the major obstacle for efficient transduction [61]. In a recent study, we demonstrated that there is a direct correlation between clathrin expression levels and baculoviral transduction efficiency in neurosphere models derived from glioblastoma patients [8]. On the other hand, other reports indicated that baculoviruses can enter several mammalian cell lines through a clathrin-independent mechanism, such as macropinocytosis [62]. It is possible that this variability in the entry of baculoviruses into mammalian cells was responsible for the different transduction efficiencies observed in this work. Regarding recombinant baculovirus generation, we used the PluriBAC system to produce occ+ recBV by homologous recombination in High FiveTM cells for use exclusively within a laboratory. However, several considerations must be addressed before these vectors can be used as inputs for future clinical translation. First, to minimize the genomic instability of recBVs and avoid the presence of potential adventitious agents, it is recommended to use rhabdovirus-free Sf9 or Sf21 cells for viral propagation. Secondly, the PluriBAC system allows obtaining transfer vectors compatible with both systems that use bacmids based on homologous recombination (occ− and occ+ bacmids), as well as systems based on transposition in bacteria (Bac-to-Bac^TM^). The flexibility of being able to choose between different bacmids is relevant in relation to biosafety. In the case of the necessity to use recBV outside the controlled environment of the laboratory, occ− viruses must be used to ensure environmental security.

BIRC6 (baculoviral IAP repeat containing 6, also known as Apollon or BRUCE in mice) is the largest member of the IAP family, with a molecular weight of approximately 530 kDa. Despite its large size, it contains only two well-characterized domains: a BIR domain at its N-terminal end, which is essential for interaction with activated caspases and the proapoptotic antagonist SMAC; and a UBC domain with E2/E3 ubiquitin ligase activity at the C-terminal end, responsible for the ubiquitination of its substrates. Little was known about the rest of its 3D configuration until 2023, when it was revealed through cryo-electron microscopy studies. We now know that BIRC6 possesses a large central functional cavity, where the recognition and catalytic activity domains converge [63,64,65]. This cavity constitutes the operational core of the protein, allowing the recognition of substrates and their subsequent ubiquitination.

In addition to its role as an apoptosis inhibitor, BIRC6 also participates in the DNA damage response by regulating proteins such as BRIT1 and activating pathways like ATM and ATR, contributing to resistance against conventional treatments and enabling the survival of damaged cells. Furthermore, it is an important regulator of autophagy, a key mechanism for metabolic adaptation and tolerance to adverse environments [25,26,27].

Taken together, the known functions of BIRC6 are integrated into several of the hallmarks of cancer, as proposed by Hanahan [66]. Given all this, it is not surprising that BIRC6 overexpression has been reported in many types of cancer, including triple-negative breast cancer, prostate, kidney, colorectal, pediatric acute myeloid leukemia, ovarian cancer, hepatocellular carcinoma, esophageal squamous cell carcinoma and non-small cell lung cancer (NSCLC) [28,29,30,31,32,67,68]. In most of these cases, BIRC6 expression has been shown to correlate with carcinogenesis, tumor progression and poor patient prognosis, as it has been demonstrated to enhance chemoresistance [33]. In the case of renal cell carcinoma, its overexpression has even been reported in cells with cancer stem cell-like properties, increasing resistance to sunitinib, a drug commonly used to treat this type of cancer [67].

In this work, we observed that BV-mediated silencing of BIRC6 increased programmed cell death (PCD), inducing 35% and 5% apoptosis in A549 and F3II cell lines, respectively, as evaluated in our in vitro experiments. These results clearly demonstrate its most well-characterized role as a member of the IAP family in NSCLC and TNBC.

Although it was originally thought that all IAPs directly blocked caspases, only XIAP is a potent and direct inhibitor of caspases 3, 7 and 9. However, its activity can be counteracted by the mitochondrial protein SMAC/DIABLO. BIRC6 is not only capable of physically restraining the activity of caspases 3 and 7, like XIAP [63,64], but it is also the only IAP known to bind SMAC/DIABLO in a manner highly similar to XIAP [63,64]. Even so, since BIRC6 is approximately 10-fold less potent than XIAP in inhibiting apoptosis, as demonstrated in extraembryonic tissue and isolated fibroblasts from mice [69], these observations raise the possibility that BIRC6 acts primarily by sequestering SMAC/DIABLO, shielding XIAP from its inhibitory effects and thereby reinforcing the suppression of apoptosis, as reflected in our results.

Furthermore, our in vivo preclinical results demonstrated that the BIRC6-silencing baculovirus was able to significantly reduce the tumor growth rate in both experimental models of breast and lung cancer. In fact, the final volume reached by tumors treated with this baculovirus was significantly smaller than that of those treated with control baculovirus. Additionally, in our mice experimental model of lung cancer, the BIRC6-silencing baculovirus significantly increased survival. Moreover, when analyzing tumor sections from both experimental models, a marked decrease in the tumor viable area (VA) and an increase in the necrotic area (NA) were observed in those animals treated with the baculovirus capable of silencing BIRC6. This increase in NA may correlate with a better response to therapy in different tumor types [47,48,49,50,70].

Taken together, these results position baculoviruses as robust antitumor gene therapy vectors that are important additions to the biomedical toolbox. In particular, we observed effects on tumor growth similar to other viral vectors currently in clinical trials, such as therapies based on both replicating and non-replicating adenoviruses [71,72,73,74].

Despite substantial advancements in research, the clinical application of gene therapy for triple-negative breast cancer (TNBC) and non-small cell lung cancer (NSCLC) remains limited by several critical challenges. One of the primary obstacles involves the optimization of both the delivery strategy and the selection of an appropriate vector for the targeted transfer of therapeutic genes into tumor cells. Viral vectors, although efficient, are frequently constrained by host immune responses, whereas non-viral systems often exhibit suboptimal transfection efficiency [53]. In this sense, the use of baculoviruses as therapeutic vectors could evade inactivation by the immune system, at least during the first inoculation, since pre-existing immunity in humans against these viruses is very low [75]. Additionally, recent progress in the development of combined baculovirus and nanoparticle-based delivery platforms [76] and engineered baculoviral vectors [77,78,79] offers promising potential to enhance the effectiveness of gene therapy.

Another significant barrier to effective gene therapy in TNBC and NSCLC is the pronounced heterogeneity of these tumors, which impedes the development of universally applicable therapeutic strategies. Both intra-tumoral and inter-tumoral genetic variability are common in TNBC and NSCLC, resulting in divergent molecular profiles among tumor cells within the same lesion as well as between patients. This heterogeneity can lead to inconsistent therapeutic responses and facilitates the emergence of resistance mechanisms, thereby reducing the overall efficacy of gene-based interventions [80,81]. The intrinsic capacity of baculoviruses to incorporate up to 38 kbp of heterologous genetic material could help solve this problem by directing therapy to multiple targets using a single vector [2,41].

## 5. Conclusions

In conclusion, considering the previous evidence on the antiapoptotic activity of BIRC6 and the effects of silencing BIRC6 expression in breast and lung cancer cells on in vitro and in vivo experimental models, we propose that baculovirus-mediated BIRC6 silencing holds strong potential as an alternative therapeutic strategy. This approach could not only enhance the efficacy of conventional treatments but also contribute to overcoming resistance mechanisms associated with apoptosis evasion in breast and lung tumors.

## Figures and Tables

**Figure 1 viruses-17-01458-f001:**
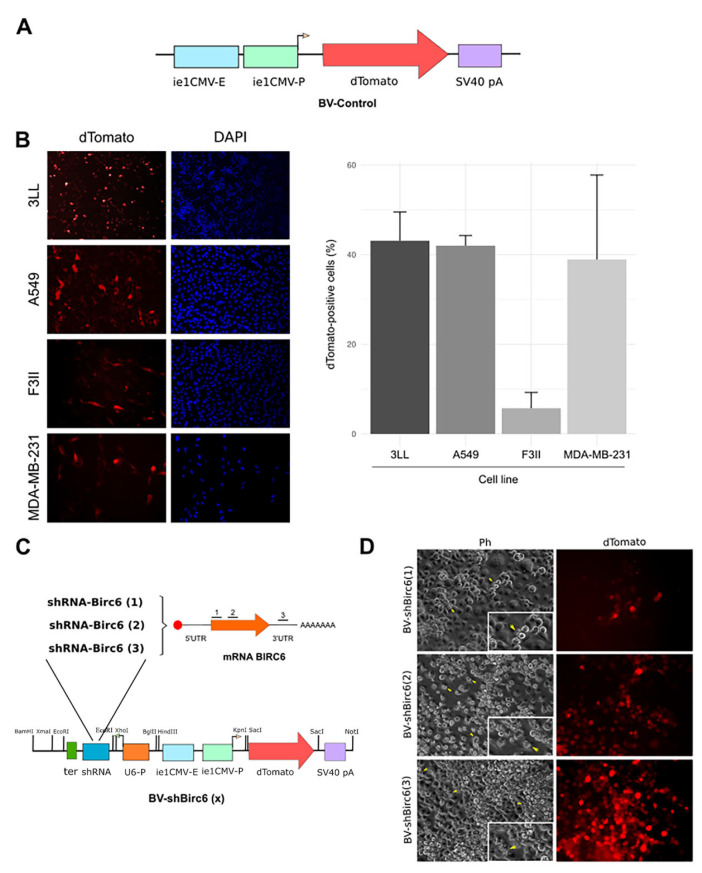
BV transduction efficiency evaluation. (**A**) Scheme of BV-Control. The reporter gene dTomato is encoded under the control of the CMV IE promoter. (**B**) Representative microphotographs of transduced 3LL, A549, F3II and MDA-MB-231 cells with 2 × 10^11^ of BV-Control for 72 h. In the left panels dTomato expression is shown, and in the right panels, total nuclei stained with DAPI is shown. Percentage of transduced cells (dTomato-positive) were plotted. Each column represents the percentage ± SD of dTomato-positive 3LL, A549, F3II and MDA-MB-231 cells, respectively (*n* ≥ 1000 cells/group). (**C**) Scheme of the silencing cassette in the BV-shBIRC6(1), (2) and (3) genome. BIRC6-specific shRNA is encoded under the U6 promoter, and the reporter gene dTomato is encoded under the CMV promoter. (**D**) Generation of recombinant baculovirus. Representative images of HighFive^TM^ insect cells co-transfected with bApGoza bacmid and the pBacPAK9–shBIRC6 transfer vector. Cells were monitored until signs of infection appeared (left) and dTomato expression was detected (right).

**Figure 2 viruses-17-01458-f002:**
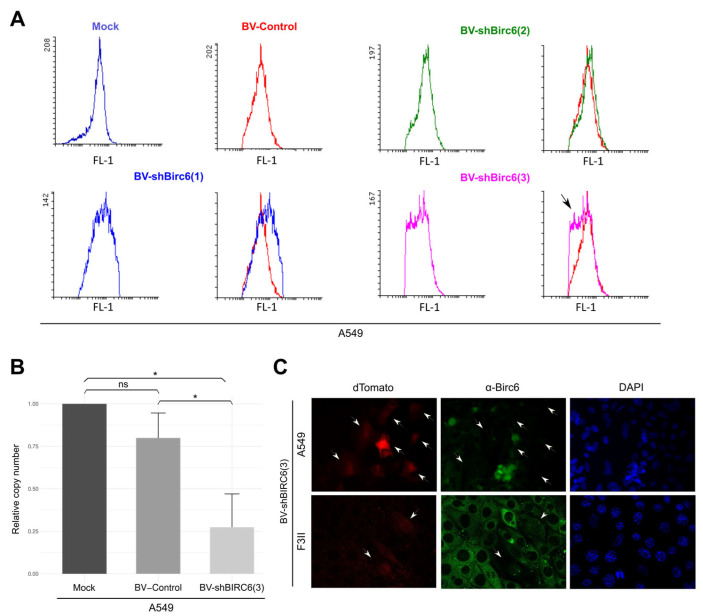
BV-shBIRC6 silencing capacity evaluation. (**A**) Evaluation of the silencing capacity of BV-shBIRC6(1), (2) and (3) on A549 transduced cells by flow cytometry (*n* = 2). (**B**) BIRC6 mRNA levels were evaluated by qPCR. Each column represents the mean ± SD of the concentration of BIRC6 RNA relative to its internal control of YWHAZ. (*n* = 3) * *p* < 0.1, ns not significant, Student’s *t*-test. (**C**) A549 or F3II cells were transduced with 2 × 10^11^ PFUs of BV-shBIRC6, as well as with BV-Control; after 72 h, BIRC6 was detected by immunofluorescence (anti-rabbit IgG- Alexa488, Vector Laboratories, 1:100). White arrow heads indicate transduced cells (dTomato+) with no or lower expression of BIRC6 (*n* = 2).

**Figure 3 viruses-17-01458-f003:**
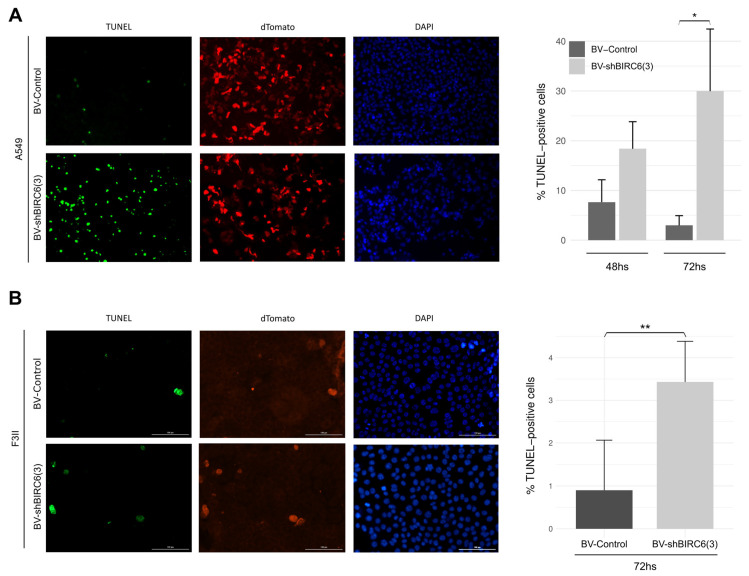
Baculovirus-mediated silencing of BIRC6-induced apoptosis. (**A**) A549 cells were treated with 2 × 10^11^ PFUs of BV-shBIRC6(3) or BV-Control for 72 h in all cases. Representative photographs of the assay measuring apoptosis levels induced by TUNEL (left), and TUNEL-positive cells quantification (right). Each column represents the percentage ± SD of TUNEL-positive A549 cells (*n* ≥ 1000 cells/group) (*n* = 2). * *p* < 0.05. χ2 test. (**B**) F3II cells were treated with 2 × 10^11^ PFUs of BV-shBIRC6(3) or BV-Control for 72 h in all cases. Representative photographs of the assay measuring apoptosis levels induced by TUNEL (left), and TUNEL-positive cells quantification (right). Each column represents the percentage ± SD of TUNEL-positive F3II cells (*n* ≥ 1000 cells/group)(*n* = 2). ** *p* < 0.01. χ2 test.

**Figure 4 viruses-17-01458-f004:**
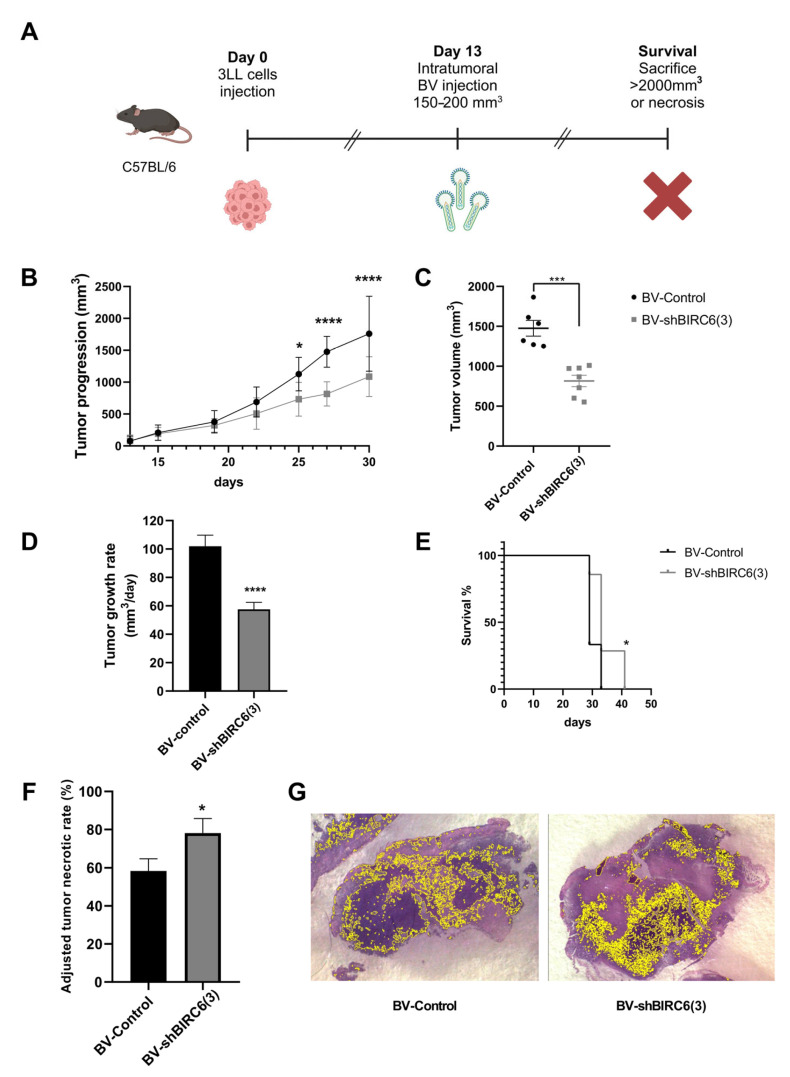
Effect of baculovirus-mediated BIRC6 gene silencing on tumor progression in syngeneic heterotopic lung cancer tumors. (**A**) Schematic representation of the workflow. C57BL/6 mice were inoculated subcutaneously with 3LL cells (3 × 10^6^ cells/mouse). When tumor volume reached 150–200mm^3^, animals were injected intratumorally (10^8^ PFUs/mouse) either with BV-Control (*n* = 6) or BV-shBIRC6(3) (*n* = 7). Created in BioRender. Pidre, M. (2025) https://BioRender.com/roe5gks. (**B**) Tumoral growth was evaluated for 30 days. Multiple regression analysis * *p* < 0.05, **** *p* < 0.0001. (**C**) At the day of sacrifice, the difference between the tumoral volume of the groups was evaluated; *** *p* < 0.0005 Welch’s *t*-test. (**D**) Tumor growth rate expressed as mm^3^/day, **** *p* < 0.0001 Welch’s *t*-test. (**E**) Kaplan–Meyer survival curves. * *p* < 0.05 versus BV-Control. Log-rank test (*n* = 6 animals per group). (**F**) % of tumor necrosis was adjusted to tumor volume variation and is expressed as adjusted tumor necrotic rates (ATNRs); * *p* < 0.05 Welch’s *t*-test. (**G**) Representative digitally stitched images of complete tumor sections from BV-Control (left) and BV-shBIRC6(3) (right)-treated mice. Yellow lines in tumor slides mark tissue regions belonging to viable malignant tissue.

**Figure 5 viruses-17-01458-f005:**
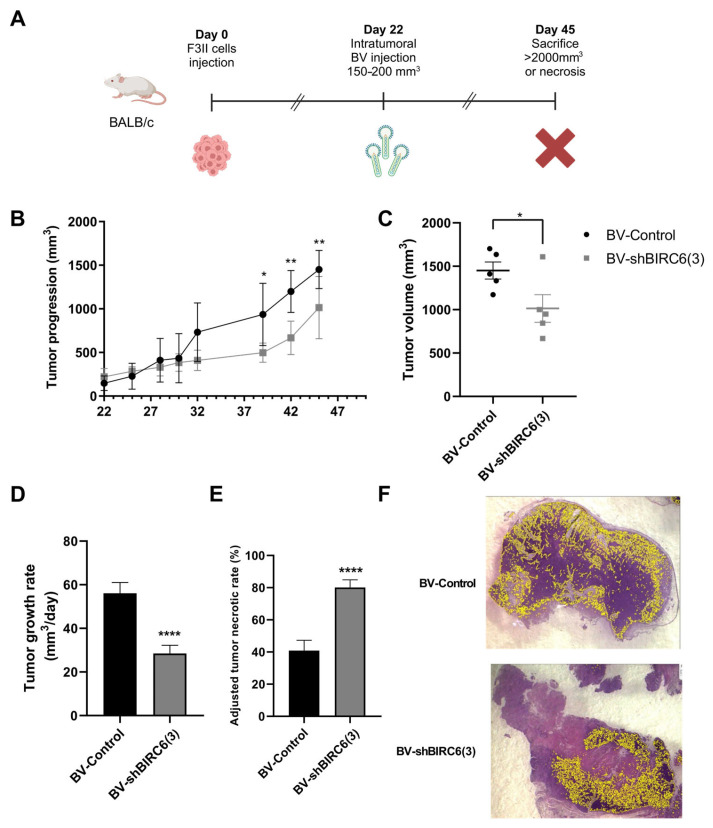
Effect of baculovirus-mediated BIRC6 gene silencing on tumor progression in syngeneic heterotopic breast cancer tumors. (**A**) Schematic representation of the workflow. BALB/c mice were inoculated subcutaneously with F3II cells (2 × 10^5^ cells/mouse). When tumor volume reached 150–200 mm^3^, animals were injected intratumorally (10^8^ PFUs/mouse) either with BV-Control or BV-shBIRC6(3) (*n* = 5 animals per group). Created in BioRender. Pidre, M. (2025) https://BioRender.com/qye9dv5. (**B**) Tumoral growth was evaluated until day 45. Multiple regression analysis * *p* < 0.05, ** *p* < 0.01. (**C**) At the day of sacrifice (day 45), the difference between tumoral volume of the groups was evaluated; * *p* < 0.05 two-tailed unpaired *t*-test. (**D**) Tumor growth rate is expressed as mm^3^/day; **** *p* < 0.0001 Welch’s *t*-test. (**E**) % of tumor necrosis in tumor samples was adjusted to tumor volume variation and is expressed as adjusted tumor necrotic rates; **** *p* < 0.0001 Welch’s *t*-test. (**F**) Representative digitally stitched images of complete tumor sections from BV-Control (up) and BV-shBIRC6(3) (down) treatment groups. Yellow lines in tumor slides mark highly basophilic tissue regions belonging to viable malignant tissue.

**Table 1 viruses-17-01458-t001:** RT-qPCR primers.

Primer	Sequence
Fw qbirc6	ATGGGCAGACAAGGCTCTCT
Rv qbirc6	TGCAGTGTTCACAATAGCCCT
Fw ywhaz	AGGAGATTACTACCGTTACTTGGC
Rv ywhaz	AGCTTCTTGGTATGCTTGTTGTG

## Data Availability

Data available upon request (mlpidre@biol.unlp.edu.ar).

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
