# Peer review of "Baculovirus-Mediated Gene Therapy: Targeting BIRC6 for Lung and Breast Cancer"

_viruses, 2025, doi:10.3390/v17111458_

Round 1
Reviewer 1 Report
Comments and Suggestions for Authors
The manuscript by Marchesini et al. describes a baculovirus gene therapy vector for BIRC6 RNA-interference, suited to increase apoptosis in lung and breast cancer cells. The authors perform a series of experiments in cultured tumor cells, demonstrating the effectiveness of the approach. Finally they perform in vivo experiments and evaluate the impact of BIRC6 shRNA BVs on tumor progression and size in two different animal models.
The manuscript is well written and the experiments are appropriately design, although occasionally additional controls could have been included. Key data (transduction efficiencies) are missing from the in vivo part, which are crucial to evaluate the effectiveness of gene delivery. As results in cultured cells show a putative decrease in BIRC6 RNA levels already with control BVs, it is extremely important to make sure that, in vivo, transduction rates of BV control and BV shBIRC6 are comparable. In absence of these data, the observed differences could also be explained by different transduction rates (e.g. incorrect viral titres or variability in gene delivery efficiencies across the two viruses).
Specific comments:
The approach of silencing BIRC6 using shRNAs and RNA-interference could have been better replaced by delivering CRISPR/Cas9 designs to target knock-out the gene. The baculovirus transgene expression is very short lived in cultured cells and in vivo. A CRISPR/Cas9 approach could have ensured a long lasting BIRC6 depletion, independent on baculovirus transgene silencing (e.g. after 72 hrs post-transduction) and examples of CRISPR/Cas9 BVs have been reported in the literature.
Figure 1D - why did the author use a HDR-baculovirus engineering approach instead of opting for a more robust Bac-to-Bac system? With the current approach, the emerging baculovirus population is likely to be a mix of loaded (dTomato) and not loaded (non fluorescent) baculovirus genomes.
Choice of shRNA sequence - was the shRNA previously validated by other lab? Why the authors have not tested multiple constructs? RNA-interference is highly sequence dependent, and different shRNA constructs targeting different portions of a gene, could significantly alter the RNA knock-down efficiency. Usually a set of 3 shRNA is tested for any new gene, unless a validated one is already available.
Why did the author package their viruses in High Five cells? Although they are ideal for recombinant protein expression, High Five are usually not adopted for BV manufacturing for gene therapy purposes due to their low titres yield. Sf9 and occasionally Sf21 are the standard in the field. This is not a criticism but I am curious if there was any specific reason for adopting this cell line.
Materials and methods
Generation of recombinant baculoviruses - …"until signs of infection (polyhedron) became apparent". Is this a feature of the PluriBAC technology or a mistake? I am not familiar with it, but broadly speaking, most bacmids for recombinant protein production are deleted for polH to avoid polyhedron production (ODVs are not formed in most rBVs). Is PluriBac different? And if so, isn't ODV production of concern for downstream applications? At the very least with excess toxicity in insect cells and diminished quality of BVs?
Figure 2 A - The decrease in BIRC6 levels by immunofluorescence are not evident in A549. Many cells have negative staining for BIRC6, and amongst those that show signal, some dTomato negative show lower levels than some dTomato positive. The chosen picture is likely not to be representative and it is not supporting very well the statement in text, or the complementary FACS and qRT-PCR data. The IF data are instead clearer for F3II, despite the lower transduction rate. For this panel I would recommend enhancing the dTomato intensity, transduced cells are barely visible.
Figure 2 B - Is the reduction in BIRC6 caused by BV-control (compared to mock) significant? Can the author perform a statistical test and discuss the results in case of identified significance? Additionally, why RNA levels were not checked in other cell lines with similar transduction rates to A549 (e.g. 3LL and MDA)?
Figure 2 C - inclusion of the mock control will increase the credibility of the data. Have these cells been gated for dTomato or the results are plotted for the total number of cells? Gating on dTomato would definitely enhance the result and, in case this has already been done, it should be clearly stated in materials and methods and figure legend.
Figure 3A/3B - Can the author include the dTomato channel to help evaluate if transduction efficiencies were similar, and if tunel+ cells are more likely to be dTomato+ for the BV-shBIRC6 rather than the control?
In vivo experiments - the data are nice, and there's a clear effect of BV-shBIRC6 in slowing down tumour progression, volume and grow rate. What is missing in my opinion are data showing that BV-control and BV-shBIRC6 have similar transduction efficiencies. Flow-cytometry of IF of tumour mass displaying dTomato expression would help confirm this. Immuno-histochemistry with anti dTomato antibodies in histological sections could also help in case the experiment cannot be repeated.
The entirety of the section 3.7 is obscure and a bit out of context for the manuscript. In my opinion it doesn't add anything to the work. The data have not been generated by the authors, and are not used to implement novel experimental approach. This kind of analysis is better suited for a review but, in my opinion, it has got no place in the current manuscript.
Reviewer 2 Report
Comments and Suggestions for Authors
The article entitled “Baculovirus-Mediated Gene Therapy: Targeting BIRC6 for Lung and Breast Cancer Treatment” is pretty interesting but need to revise major things. After that can be considered too publishable in MDPI “Viruses” journal. The following comments should be addressed.
Comments-1: The title of the article should be AI guided notation free “Baculovirus-Mediated” I think it should be “Baculovirus Mediated”. The title should be consistent with the main contain of the paper. Also must be used proper logics of words such as “Therapy” and “Treatment” both carry same objects. The title can be like “Baculovirus Mediated Gene Therapy: Targeting BIRC6 for Lung and Breast Cancer”. Abstract must be contain main theme of the research not background of the research, it contain almost 50% overall presentation. In the introduction section reference are not properly placed. Also, Introduction is to short it should be discussed elaborately previous reports related to BIRC6 for Lung and Breast Cancer Treatment.
Comments-2: Abstract and conclusion to similar it should be extensively discussed. Overall data presentation are well organized and discussed clearly to supports the research idea. Way of presentation of the figure is not appropriates such as Figures 6A upper and lower panels, it can be Figure 6A and 6B. In figure 5B panel in the Tumor volume should be specify with proper figure number.
Also take a special attention on AI guided notation throughout the paper “insect-specific”, “BV-Control”, “post-inoculation” and “Genotype-Tissue” and so on.

Round 2
Reviewer 1 Report
Comments and Suggestions for Authors
I thank the authors for taking their time in replying to my comments.
The inability of the authors to provide basic additional experiments and controls in some cases (e.g. dTomato expression in tunel pictures) is occasionally motivated by lack of funding which, in my opinion and based on my knowledge of the costs for requested experiments, is surely not the primary reason.
While I still think that the manuscript is publishable in principle, I would have loved to see a more proactive experimental approach towards answering some of the comments.
Some of the justifications for the experimental approaches chosen, even in this revised version of the manuscript, are sometimes misleading for readers (e.g. usage CRISPR vs RNAi in clinical trials).
Additionally, the use of occ+ and occ- viruses complicate the matters as it is unclear 1) why occ+ viruses were used in the context of BV production for gene delivery to mammalian cells, 2) what is the risk of using occ+ viruses for such purposes (including the risk for environmental spread to insect populations), 3) when and for which experiments occ+ or occ- viruses were used.
Unfortunately, taking together these consideration, I am not 100% convinced of the robustness of the work (e.g. some experiments appear to have been made only once), or its validity for in vivo delivery (lack of proof of successful transduction in vivo, besides the effects on tumour growth).
I also notice that, at least to my knowledge, I am the only reviewer for this work so far. I would strongly recommend to the editor to seek at additional editors who might agree or not with my views. It is not standard practice to have a paper examined by only one reviewer and surely.
Below, and extract of previous correspondence (the points not included here have been addressed in full). A = authors, R = reviewer, R1 = round 1 comment, R2 = round 2 comment.
Specific comments
Comment 1
A: The reviewer's point is very interesting. The versatility of baculoviruses would allow the use of CRISPR/Cas9 technology to produce gene knockouts, as demonstrated in previous reports [1,2]. However, despite advances in gene editing, there is still a long way to go before it can be translated into clinical applications. Taking this into account, and the fact that clinical trials involving gene silencing using viral vectors are currently underway, we chose this strategy, which could bring us closer to that ultimate goal. Attending this well-founded concern of the reviewer, we have incorporated this debate into the Discussion section (line 534).
R: This explanation, and the amended text at line 534 of the manuscript, downplays the clinical trials involving CRISPR, incorrectly suggesting that the technology is not as mature as RNAi. Indeed, the clinical trials landscape seems to indicate quite the opposite. I would recommend the authors not to include this as an explanation as to why they opted for RNAi vs CRISPR, as the logic could mislead readers. The authors could instead indicate, for instance, that they preferred RNAi to avoid the generation of additional DNA damage in cell lines and models (tumoral) in which DNA repair pathways and DNA integrity checkpoints are often mutated.
Comment 2
A: Our system of producing transfer vectors for recombinant baculovirus generation (PluriBAC) is compatible with both HDR and Bac-to-Bac systems. Both are susceptible to producing wt genome contamination.
R: The Bac-to-Bac system does not produce wt genomes. The Tn7 recombination is ensured before bacmid transfection, as to provide isoclonal bacmids containing the correct gene of interest. With HDR-based systems instead, wt bacmids are co-trasfected with the gene of interest, which is then integrated in a given bacmid locus. Despite inclusion of complementation technologies, the HR-based systems are at a greater risks of amplifying non-recombined bacmids (which can be complemented in trans at very early stage during viral amplification due to co-transfection or co-infection of functional and defective (e.g. ORF 1620 KO) genomes.
Comment 3
A: Regarding the different systems that can be used to generate baculoviral vectors to transduce mammalian cells, it is also important to note that Bac-to-Bac has multiple resistances to antimicrobial agents that would also be undesirable in the context of biomedical applications. That is why we ideally seek a system free of antimicrobial selection markers. To avoid wt genome contamination, we use the occ+/occ- bacmids derived from the works of Je, et al [3–5]. These bacmids are defective in the essential gene ORF 1629, significantly reducing wt genome contamination. However, to ensure the purity of recombinant baculoviruses, we worked with fewer rounds of infection, and we compared titers obtained by plaque assays counting both fluorescent and polyhedron producing plaques.
R: Bac-to-Bac system, in its commercially available form (DH10Bac E.Coli Invitrogen) carries only one antibiotic resistance marker (kanamycin in the mini-F replication origin). This is common to the Pluribac system (unrecombined), which is transfected into insect cells. Bac-to-Bac requires one additional resistance marker (often gentamycin) for selection of recombinant clones. The advantages of reducing the antimicrobial selection markers are thus not immediately clear, as they are not expressed in insect nor mammalian cells (bacterial promoters). More concerning, is the use of occ+ viruses for the generation of delivery vehicles for mammalian cells. While ODV are unlikely to be co-purified or transferred to mammalian cells, the use of occ+ viruses is of particular concern for environmental containment, since recombined ODV (containing the rBV genome) can easily spread to insect populations in the wild, something that cannot happen with occ- virions.
Comment 4
A: The reviewer is correct in his/her observation. In fact, we tested three different sequences as potential shRNAs, observing silencing activity only in the recombinant baculovirus carrying one of them. We now included the details and flow cytometry validation of these sequences in a Supplementary Figure and mentioned it in the Methods section (line 122).
R: The Supplementary figure supports a preliminary assessment of different shRNA constructs. The new figure is however depleted of statistical information, and only shows one replicate. How many times was the experiment repeated? Please also note that control and BV-shBIRC6 histograms are the same depicted in Figure 2. I understand that this was probably the same experiment, but data should in theory not be duplicated across the manuscript. I would therefore recommend moving the preliminary shRNA study in Figure 2 (main text) after adding the requested statistical analysis or replicates information.
A: We thank and address the reviewer's observations. Matindoost and collaborators [6] have shown that baculovirus titers depend on the cell line and culture medium used. In fact, they observed a higher yield when AcMNPV is propagated in Sf9 cells, compared to Tni cells (High FiveTM). However, the assays carried out by Matindoost and colleagues did not include Grace’s culture medium, which is typically used in our lab. Based on our experience, we have observed similar behavior in both cell lines with respect to BV titer yield. While Wilde and collaborators [7] also showed that it is possible to achieve higher baculovirus titers using Sf9 cells in comparison to Tni cells, they have reported an unique experimental condition (MOI=1, and harvesting 6 days post infection), which differs from our usual protocol.
R: I thank the authors for their explanation.
Comment 5
A: Actually we have two different bacmids compatible with the generation of recAcMNPV by homologous recombination in insect cells. Both derive from the works of Je et. al. The generation of the occ+ bacmid was reported in "Generation of baculovirus expression vector using defective Autographa californica nuclear polyhedrosis virus genome maintained in Escherichia coli for Occ+ virus production." [4] and that was the bacmid used for the generation of our recombinant BV. While it is true that the use of an occ+ bacmid could affect the yield of budding virions due to their occlusion, under the conditions and infection times with which we worked, we did not have any complications in obtaining the necessary titers to perform the assays.
R: The use of occ+ viruses in gene therapy has never been reported (to my knowledge). It is also unclear why (and where) the occ+ or occ- viruses where used. I would strongly recommend to indicate this clearly in material and methods, and to indicate additionally for which experiments the occ+ and occ- viruses where used, respectively (perhaps in figure legends). This is an important point for readers who might want to reproduce these results in the future, and I would re-instate this in discussion, alongside any consideration that the authors might want to add.
Comment 6
R-R1: Figure 2 B - Is the reduction in BIRC6 caused by BV-control (compared to mock) significant? Can the author perform a statistical test and discuss the results in case of identified significance?
A: There are no significant differences between mock and BV-control transduced cells BIRC6 mRNA levels. Statistical reference was added to panel 2B.
R-R2: Thanks for including statistical information. Please also amend figure legends with the number of replicates, as this is not immediately clear from the figure. Also amend the figure legends with replicate number information (n=) for the rest of the figures in the manuscript. Apologies for not including this comment earlier.
Comment 7
A: Unfortunately, due to budget limitations imposed by the Argentinean government, we could have validated the shRNA effect only on the A549 cell line.
R: While I sympathize with the lack of resources to carry out additional experiments, the lack of funds surely cannot be called as a primary cause for not carrying out the experiment. For each cell line the cost would have surely been negligible for either FACS or qRT-PCR given the fact that reagents (e.g. antibody for FACS) were in the lab.
Comment 8
R-R1: Figure 3A/3B - Can the author include the dTomato channel to help evaluate if transduction efficiencies were similar, and if tunel+ cells are more likely to be dTomato+ for the BV-shBIRC6 rather than the control?
A: In this work, we show that BV-shBIRC6 induces 30% apoptosis in A549 cells and 3% in F3II cells, considering the total cell population. We did not normalize to transduced (dTomato-positive) cells because the aim was to assess the global effect on apoptosis evasion, a cancer hallmark, reflecting the tumor context. BV transduction efficiency was nearly 40% in A549 and less than 10% in F3II. Therefore, the observed apoptosis may represent either a direct effect of BV on tumor cells or a bystander effect.
R-R2: I totally get the correlation between transduction and silencing efficiencies. However I find it difficult to believe that a researcher went all the way to the microscope to take TUNEL images and did not acquire a dTomato channel. Even in this event, surely the IF slides are still available and the images could be easily reacquired.
Comment 9
R-R1: In vivo experiments - the data are nice, and there's a clear effect of BV-shBIRC6 in slowing down tumour progression, volume and grow rate. What is missing in my opinion are data showing that BV-control and BV-shBIRC6 have similar transduction efficiencies. Flow-cytometry of IF of tumour mass displaying dTomato expression would help confirm this. Immuno-histochemistry with anti dTomato antibodies in histological sections could also help in case the experiment cannot be repeated.
A: The reviewer's observations are very accurate and it would be really desirable to have the expression of dTomato at the end of the in vivo trial. However, according to our experience and previous reports, we have been able to observe baculovirus-delivered dTomato expression up to 14 days post-intratumoral inoculation in immunocompetent mice and up to 21 days in CNS models [8,9]. These limitations of the delivery system are the reason why we did not look for the presence of dTomato in the tumor sections. On the other hand, to ensure comparable transduction efficiencies between BV-Control and BV-shBIRC6, we titrated the viral inocula by triplicate and verified their transduction efficiency in vitro before inoculating them into the animals. Likewise, BV-Control (Ac-dTomato) was used as a control in different previous reports from our research group without showing any type of antitumor effect in other experimental models [8,10].
R-R2: I am familiar with the silencing observed with BVs both in vivo and in vitro. Despite the functional characterisation and its effects on tumour size, in absence of dTomato expression data, no evidence is provided about BVs successful transduction. Also if BVs can only produce dTomato expression until 21 days pt, why the effects on tumour growth are only becoming obvious after this period? Is dTomato silenced while the shRNA keeps being expressed?
Comment 10
R-R1: The entirety of the section 3.7 is obscure and a bit out of context for the manuscript. In my opinion it doesn't add anything to the work. The data have not been generated by the authors, and are not used to implement novel experimental approach. This kind of analysis is better suited for a review but, in my opinion, it has got no place in the current manuscript.
A: The Cancer Genome Atlas (TCGA), a joint initiative of the NCI and NHGRI, characterized more than 20,000 tumors and matched normal samples across 33 cancer types. By integrating clinical, genomic, epigenomic, transcriptomic, and proteomic analyses, TCGA generated comprehensive, publicly accessible datasets that have advanced understanding of cancer biology, facilitated discovery of therapeutic targets, and enabled precision oncology.
Using TCGA datasets, we can analyze our molecular target expression in hundreds of patients and perform correlation analyses between our molecular target and other genes relevant to cancer physiology, among others. This type of analysis allows us to better understand the role of our molecular target in the tumor types studied, not only in experimental models but also in samples from patient biopsies. That is why we included this type of in silico analysis in our study.
In response to the reviewer's concerns, we included a paragraph explaining the rationale for this type of analysis in the results section and its contribution to our work (line 472).
We hope to meet the reviewer's expectations with our response and the addition of the rationale for the analysis. Otherwise, we might consider moving this section to the supplementary material.
R-R2: I appreciate the effort, but the purpose of the section is still obscure to me in the context of the current manuscript. It is an attempt to a posteriori attribute importance to the chosen target. I will not question further the decision of the authors, but I unfortunately remain of the idea that this section is entirely out of place and unconnected to the rest of the manuscript (main or supplementary).
Reviewer 2 Report
Comments and Suggestions for Authors
Author described all the reviewer questions in the current version. It can be accepted without further revision.
Author Response
We thank the Reviewer for recommending our manuscript for further publication.